# Goal Directedness, Chemical Organizations, and Cybernetic Mechanisms

**DOI:** 10.3390/e23081039

**Published:** 2021-08-12

**Authors:** Evo Busseniers, Tomas Veloz, Francis Heylighen

**Affiliations:** 1Centre Leo Apostel for Interdisciplinary Studies, Vrije Universiteit Brussel, B-1160 Brussels, Belgium; evo_busseniers@hotmail.com (E.B.); fheyligh@vub.ac.be (F.H.); 2Fundacion para el Desarrollo Interdisciplinario de la Ciencia, la Tecnologia y las Artes-DICTA, Santiago 8330307, Chile; 3Facultad de Ciencias para la Vida, Universidad Andres Bello, Santiago 8370146, Chile

**Keywords:** chemical organization theory, goal-directedness, cybernetic mechanisms

## Abstract

In this article, we attempt at developing a scenario for the self-organization of goal-directed systems out of networks of (chemical) reactions. Related scenarios have been proposed to explain the origin of life starting from autocatalytic sets, but these sets tend to be too unstable and dependent on their environment to maintain. We apply instead a framework called Chemical Organization Theory (COT), which shows mathematically under which conditions reaction networks are able to form self-maintaining, autopoietic organizations. We introduce the concepts of perturbation, action, and goal based on an operationalization of the notion of change developed within COT. Next, we incorporate the latter with notions native to the theory of cybernetics aimed to explain goal directedness: reference levels and negative feedback among others. To test and refine these theoretical results, we present some examples that illustrate our approach. We finally discuss how this could result in a realistic, step-by-step scenario for the evolution of goal directedness, thus providing a theoretical solution to the age-old question of the origins of purpose.

## 1. Introduction

The notion of goal directedness or purpose has long been considered to be outside the realm of science [1]. Physical science assumes that effects are fully determined by their causes, which lie in the past. Therefore, it does not seem possible for a goal, which lies in the future, to affect phenomena here and now. Yet, the world is full of systems, such as bacteria, people, or organizations, that behave as if they are striving to achieve some as yet distant goal state. That means that whatever their initial state (cause), they will act so as to reach this particular end state, thus making it appear as if it is this end state and not the initial state that determines their course of action. Moreover, this end state is not a natural equilibrium, such as a ball coming to rest at the bottom of a pit, but a far-from-equilibrium state that requires active intervention and therefore a continuing mobilization of energy to achieve and maintain.

A solution to this paradox, first suggested in a classic paper entitled “Behavior, Purpose and Teleology” [2], was proposed by the theory of cybernetics. Cybernetics introduced the notion of circular causality to explain how an end state can affect an initial state [3]. Although apparently paradoxical, circular causality can be implemented in the simplest case by feeding the output (effect) of a process back to its input (cause). If the feedback is negative, this will suppress any deviations from a “reference” state. This guarantees that, whatever disturbances may drive the causal conditions away from this goal, their effects will be neutralized so that the goal state is dependably achieved. This can be illustrated by a heat-seeking missile, which continuously monitors the difference between its position and the one of the heat source it is targeting, while adjusting its course so as to reduce that difference and thus eventually hit the target.

Early cybernetics developed a mathematical theory inspired by such control technologies, which included thermostats, servomechanisms, and anti-aircraft artillery. However, these systems were artificial, meaning that their goals were imposed by their human designers. Cybernetics later extended its reach to describe naturally goal-directed systems, such as organisms, brains, and social systems [4,5]. Here, the goal is not determined from the outside, but intrinsic to the system. These systems are autonomous or self-steering [6], and thus can be said to exhibit agency. For systems that are the product of evolution, the implicit goal is survival and reproduction. The reason is that systems that did not effectively target and achieve this goal have been eliminated by natural selection. However, to reach this overall goal, evolved systems need to aim at several more concrete, subsidiary goals, such as seeking food when hungry, warmth when cold, or safety when threatened.

Applied to organisms, the cybernetic notion of circular causation inspired what is perhaps the most general definition of life: autopoiesis [7,8]. Autopoiesis can be seen as a generalization of the notions of metabolism and reproduction that characterize living cells. An autopoietic system is a network of processes that continuously (re)produces its own components, so as to ensure that its organization survives both the wear and tear of entropy and any external disturbances threatening its integrity. Its implicit goal is self-maintenance. Thus, it exemplifies a structure with emergent purpose: its components and processes are merely simple causal mechanisms, yet together they form an autonomous “agent” or “self”, i.e., an organizationally (but not thermodynamically) closed whole that will act so as to ensure its continued existence. Here, the simple feedback loop has been expanded into a complex network of synergetic relations that make the different processes support each other in the service of the emergent goal. While autopoietic systems are goal-directed, the theory does not explain how such living systems could have evolved out of earlier abiotic systems that lacked this feature. Independently of autopoietic and cybernetic theory, many scenarios have been proposed for the origin of life. These fall roughly into three categories, depending on whether they put the priority on the emergence of a metabolism, a cell membrane, or a replicator such as RNA. While much progress has been made in understanding specific physical and chemical conditions that may give rise to certain of these elements, the overall picture remains obscure. The problem is that none of these scenarios explains how such abiotic components could have developed purposeful action or agency (Deacon, 2011). That seems intrinsically difficult when the starting elements are linear causal mechanisms and passive physical objects, such as molecules or membranes, rather than actively self-maintaining organizations.

From the proposed scenarios, the one that comes closest to the circular causality demanded by cybernetics is the self-organization of an autocatalytic cycle [9,10]. However, this scenario has a fundamental shortcoming. Autocatalysis is by definition a form of positive feedback: the more there is of a catalyst, the more will be produced. This runaway growth makes the cycle intrinsically unstable: instead of coming to a “desired” concentration (goal), its reserve of catalysts grows exponentially, until the “food” molecules it needs for further production are exhausted. Unless there is a dependable outside source of food, this entails the interruption of the cycle and therefore the death of the incipient “metabolism” [1]. Another intrinsic weakness of complex autocatalytic cycles (such as “hypercycles”) is the “error catastrophe”: if at some stage a wrong catalyst is introduced, this error cannot be corrected, and is likely to snowball into errors producing further errors, until the cycle completely breaks down.

Our aim is to propose the emergence of target-oriented behaviors, providing a rigorous conceptualization of such “agentic” behaviors and their evolution. More generally, our project explores how a better understanding of the mechanisms of self-organization and adaptation that govern the universe may help us to discover meaning and purpose in life. We propose that human behavior too may be conceived as the course of action performed by an autonomous, autopoietic agent, striving to maintain and develop its identity within a complex, challenging environment, by using various cybernetic and resilience strategies.

The new formalism of Chemical Organization Theory (COT) shows mathematically under which conditions reaction networks self-organize into self-maintaining, autopoietic organizations [11], and thus can be understood as basic structures in which goal-oriented behaviour can emerge [12].

This theory is complementary to autocatalytic sets [13] and to other formalisms relevant in reaction networks such as deficiency theory [14] and Petri nets [15]. However in this article, we will focus on explaining some basic notions to frame the representation of goal-directed systems using reaction networks, and particularly COT.

We first describe our general perspective to model goals, next we introduce the elements of COT that allow for representing such perspective, later we present some examples and we end with a discussion of results and future work.

## 2. Goal Directedness

The main question we are interested in is how goal-directed behaviour emerges. To answer this, we should first clarify what we understand by goal directedness.

Our general perspective is that of Beer: “the purpose of the system is what a system does” [16]. Thus, if a system is doing actions following a certain pattern which directs it in a certain way, we consider such way its goal. The system should move towards the direction defining its goal under a variety of circumstances. The latter is known as *equifinality*: no matter the conditions one start in, the system will always achieve the same result [17,18].

Equifinality does not necessarily refer to an end-state of the dynamics, as most systems are far-from-equilibrium. However, systems “actively” move to, as if they were seeking for, such end-state [19]. Therefore, what we should have for goal-directedness, is that the direction is chosen, or appears to be chosen, by the system, and is not simply an external cause–effect mechanism. We use the language of reaction networks, as it is possible to describe systems that cause themselves by circular causality: the output (effect) can process back to the input (cause) [3,12,20].

As an example of the difference between external cause–effect and goal directedness, consider the wind as an outside influence. An example of an external cause–effect is a leaf that is blown by the wind, and goes in whatever direction the wind blows. By contrast, sailing is a goal-directed behaviour: the direction the boat is going does not depend on the wind, it can even go against the wind (by zigzagging). Obviously sailing is a behaviour not isolated from the environment, it is using the wind to move into the wanted direction. While it does not need to alter its environment to reach its goal.

Therefore, we can identify three aspects related to goal-directed-behaviour. There are **perturbations**, by which we understand all influences on the system, both the typical dynamical rules of the environment of the system as well as the accidental events that change the state or the dynamical rules of the system. The system reacts to these perturbations by certain **actions**. Again, actions can be the direct consequence of the dynamical rules of the system and its environment, or might accidentally be enabled or disabled by the perturbations. Actions have certain effects. While both the perturbation and actions can have a huge variety, for a system to be goal-directed, the actions should follow a certain pattern that is observable, something that must remain stable enough to be named “a goal”. We call whatever this something is in a certain system its **goal**. In Figure 1 we illustrate how goals, perturbations, actions, and feedback mechanisms are related. The latter can be an end-state in the simplest case, but it can also be a stable pattern in time, where even if the system is constantly changing, the pattern remains present. For example, remaining at a certain part of the state space. In general, a goal entails a reduction in entropy: while before the goal there is no a priori preference for a certain region of the phase space, when there is a goal, certain states will be more probable, and thus the entropy will be lower [21].

Cybernetic mechanisms allow to explain some of the goal-directed behaviors [22]. For example, the simplest case of a (direction) parameter that stays stable can be described by a **negative feedback** [3]. This is when perturbations are counteracted, so that whenever the parameter is higher than a certain reference value, an action occurs so that this parameter lowers, while if the parameter is lower than the reference value, it will get increased. In this way, the parameter will stay at a certain level. A negative feedback is fairly simple to emerge naturally. Predator–prey dynamics, for example, follow a negative feedback: if the number of predators grows, the number of prey will decrease, lowering the amount of predators.

The opposite of negative feedback is **positive feedback**. This is when the more there is of something, the more is produced, and the less there is, the less there gets. Positive feedback creates instability, while negative feedback creates stability.

To have more complex goals and goals interacting with each other, a negative feedback can interact with other negative feedbacks and other mechanisms, for example, there can be a mechanism influencing the specific reference value to which a negative feedback moves [23]. While a negative feedback brings goal-directed behaviour (as something is kept stable), its shortcoming is that it can only react after the perturbation has happened. Two mechanisms exist that do not have this drawback: buffering and feedforward control. Buffering means the absorption of fluctuations by accumulating reserves of resources. Feedforward control means acting before a perturbation can drive the system away from its goals [3].

We are interested in how goals can emerge, thus how starting from purely natural phenomena, we would call an action to be a “decision”. We can hence not simply assume there is an agent triggering certain actions, but are interested in how such an agent can turn up.

This evolution of increasingly more complex goals can happen because simpler negative feedbacks, goals, and other mechanisms interact with each other. Evolution is often possible because of variation and selection: there is a variety of cases, and those that are best in surviving, will survive and thus be selected (which framed in this way is a tautology). However, we can not simply state that the main, overall goal is always survival: there are different equally valuable strategies to get there, several trade-offs, different ways to measure what we understand by “survival”, and sometimes systems have goals that go against or are independent of their survival. What we can say is that a system usually creates a coherence between different goals, goals become synergistic so that they can mutually foster each other. This is in line with the definition of goal directedness by [17] as coherent dynamical coherence. Dynamical coherence is a similar concept to equifinality: the system moves to the same result under a variety of circumstances. However, to be goal-directed, there should be a coherence with the rest of the system’s dynamics, different parts should work together.

In our approach, a goal-directed system will bootstrap itself: because of the cyclical nature, it can create itself, it can decide its own direction, produce itself, and create new goals. The principle of self-creation and self-maintenance is called autopoiesis [8].

To see how exactly goal-directed behaviour can emerge, we need a formalism that can represent cause–effect basic processes, so-called reactions, and how these reactions self-organize into something that appears to be goal-directed [19]. More precisely, we want to look at how reactions can interact in such a way that goal-directed behavior emerges, that they mutually influence each other such that they are self-maintaining. Chemical organization theory provides such formalism, which will be introduced in the next section.

## 3. Modeling Goal-Oriented Systems Using Reaction Networks

Reaction networks are at the core of geo-biochemical processes, and thus most evolutionary processes known to occur in our planet can be reduced to changes in the structure and operation of reaction networks [24]. From an abstract perspective, a reaction network is simply a collection of reaction rules that describe how certain collections of entities transform into other collections of entities. Therefore, the dynamics of a collection $M=\{s_{1},\ldots ,s_{m}\}$ of *species* that can react with each other according to a set $R=\{r_{1},\ldots ,r_{n}\}$ of *n reactions* correspond to specifying the influence that different possible configurations of species in M imply in the happening of reactions in R.

For a given reaction r∈R, the species supp(r) required to trigger *r*, are called the *support* or *reactants* of *r*, and the species prod(r) to be created by this transformation are called *products*. Together, the set of species and the set of reactions are called the *reaction network* (M,R).

In general, some reactions in R might occur more often than others. A particular specification of the occurrence of reactions within the reaction network is called *reaction process*, or simply *process*. In reaction network modeling, v is usually called *flux* vector, and it is normally computed directly from the state x(t)=(x[1](t),…,x[m](t)), where x[i](t) corresponds to the concentration of $s_{i}$ in the reaction network at time *t*, i=1,…,m. We introduce the term process instead of flux to remark that our aim lies beyond the modeling of biochemical systems.

### 3.1. Closed and Semi-Self-Maintaining Subnetworks

Note that for any set of species X⊆M, there is a maximal set of reactions $R_{X}$, defined as the set of all reactions whose reactants are in *X*. Thus, each set *X* induces a *subnetwork*$\left(X,R_{X}\right)$.

In general, the reactions in $R_{X}$ might produce species which are not in *X*. Consider for example the set X={a,b} and the reaction {a+b→c}. Clearly, the dynamical operation of the reaction network starting in a state where only species in *X* are present, will at some point have to incorporate also *c* in its dynamics. We introduce the notion of *closed* set to denote those sets of species whose dynamical operation is complete, i.e., where $R_{X}$ does not produce novel species to *X*. Formally, *X* is closed if and only if $\cup_{r\in R_{X}}\mathrm{prod}\left(r\right)\subseteq X$. Moreover, for any set *Y* we denote by $G_{CL}\left(Y\right)$ the closure of *Y* as the smallest closed set containing *Y*.

**Lemma** **1.**
*X is closed if and only if $G_{CL}\left(X\right)=X$.*


Given any set, we can obtain its closure by subsequently adding the products of reactions triggered by the set. Closure, thus, confines all the species we should take into consideration; it defines the behavioral scope. A set will naturally evolve to its closure because additional species will be added as the product of reactions triggered, which will trigger more reactions and thus add more species.

However, some closed sets might evolve into smaller sets. For example, consider the reactions
$$r_{1}=a\to b,\;r_{2}=b\to c,\;r_{3}=c\to b.$$
In this case, we have that {a,b,c} is a closed set, but as *a* is consumed by $r_{1}$ but not produced by any reaction, we can guess that any initial condition which begins having positive concentration for the three species, will in the long run end up in the set X={b,c}, which is not only closed but also *semi-self-maintaining*, meaning that $R_{X}$ is able to produce all the species that it consumes by its reactions. Formally, *X* is semi-self-maintaining if and only if $\cup_{r\in R_{X}}\mathrm{supp}\left(r\right)\subseteq \cup_{r\in R_{X}}\mathrm{prod}\left(r\right)$.

This can be seen as a next step in the evolution, where species are deleted when they cannot be maintained. In this way, the reaction network becomes more and more stable. While the system is in constant flux, as specific instances are constantly consumed and produced, there is an invariance emerging.

This evolution can be seen as an illustration of the more general *principle of self-organization* [20,22,25]. This states that any dynamic system will eventually end up in an attractor: a region the system can enter but not leave.

Note that semi-self-maintenance is a structural form of quantitative self-maintenance. Consider, for example, the reactions
$$r_{1}=2a\to b,\;r_{2}=b\to a.$$

This reaction network is closed and semi-self-maintaining, but its operation will tend to reduce the total amount of species because in order to produce one species of type *a* we need to consume one species of type *b* which in turn needs two species of kind *a* to be produced. Thus, for every two species of type *a* consumed, only one can be produced. In order to account for a quantitative notion of *self-maintenance*, we need to specify how productive processes occur in a reaction network.

A reaction $r_{i}$ is now represented by
(1)$$r_{i}=a_{i1}s_{1}+\ldots a_{im}s_{m}\to b_{i1}s_{1}+\ldots b_{im}s_{m}$$
with $a_{ij}$, and $b_{ij}\in N_{0}$, and i=1,…,n.

The number $a_{ij}\in N_{0}$ denotes the number of reactants of type $s_{j}$ of the *i*-th reaction. Together, these numbers form a *reactant matrix* $A\in N_{0}^{n\times m}$. Analogously, the number $b_{ij}$ denominates the number of products of type $s_{j}$ of the *i*-th reaction. Together, these numbers form a *product matrix* $B\in N_{0}^{n\times m}$. From here, we can encode the way in which species are consumed and produced by the reactions in the stoichiometric matrix S=B−A.

As the stoichiometric description counts the amount of each type of species involved in the reactions, processes can be extended to specify the number $v_{i}\in N_{0}$ of times that each reaction $r_{i}$ occurs. Therefore, a process corresponds to a vector v=(v[1],…,v[n]), and thus a process v can be applied to *X* only if $r_{i}\notin R_{X}$ implies that v[i]=0.

Note that the application of a process v to *X* in the state x will lead to a new state $x_{v}$ that can be computed by the following equation:(2)$$x_{v}=x+Sv.$$

Namely, the new state $x_{v}$ is equal to the previous state x plus a production vector Sv whose coordinates indicate the amount of increase or decrease of each species after the process is applied.

The latter allows to define self-maintaining reaction networks as those in which there are productive processes that quantitative preserve the reaction network.

**Definition** **1.**
*X is self-maintaining if and only if there exists v such that v[i]>0 for every $r_{i}\in R_{X}$, and $x_{v}\left[j\right]\geq x\left[j\right]$, j=1,…,m.*


For self-maintaining sets, we can find processes such that every reaction in $R_{X}$ occurs, and the result of the process does not lead to the consumption of any species. Therefore, self-maintaining sets entail the parts of the reaction network where self-sustainable processes, at a quantitative level of description, can occur.

We can look to the goal directedness from the perspective of one specific species, where its goal is to remain. The reactions consuming this species can be seen as perturbations, while the reactions producing that species can be seen as actions done to still aim for the goal. If in the whole there is more produced than consumed of that species, we can consider that the goal of that species can be reached.

The goal of one species interact with the goals of other species, sometimes in a synergetic way (one species triggering the production of another species), but often these goals are conflicting (the consumption of one species is needed for the production of another). However, if there is self-maintenance, there has been some mutual adaptation so that all the goals of the species can be reached, i.e., all the species can remain present. The different goals have become coordinated, they work together, and thus a coherence between them emerges.

Note that if *X* is self-maintaining, then *X* is semi-self-maintaining, while the opposite is not always true.

### 3.2. Goal-Oriented Systems Are Organizations

Chemical Organization Theory combines the notions defined above to introduce the concept of *organization* as a closed and self-maintaining set [11]. By combining these two requirements, organizations represent a structural footprint of potentially stable dynamics. Namely, a closed set of species entails a subnetwork whose processes do not produce new species, and within this closed dynamics, there are processes that allow quantitative self-production of the species. Therefore, as long as self-maintaining processes occur, the subnetwork $\left(X,R_{X}\right)$ will be preserved in time.

In general the time-evolution of a reaction network is represented by a (continuous or discrete) system of equations, known as a *reaction system* in which the state x(t) together with some parameters determine the process v(t,x(t)) to be applied to the reaction network, and which in turn determines the infinitesimal or discrete change in time Sv(t,x(t))).

As the equations required to compute the time-evolution of a reaction network are, even for moderately complex cases, nonlinear and highly coupled, it is complicated to know what is the future of a reaction network given its initial conditions. However, the notion of organization helps to obtain crucial information regarding the dynamics. Before, we need to introduce some notions:

**Definition** **2.**
*Let $P\left(M\right)$ be the power set of M and*
(3)$$\begin{array}{c} \phi \left(t\right):R_{\geq 0}^{m}\to P\left(M\right),\;x\left(t\right)\mapsto \phi \left(x(t)\right)\equiv \left\{s_{i}\in M:\;x_{i}\left(t\right)>0\right\}. \end{array}$$

*For a state $x\left(t\right)\in R_{\geq 0}^{m}$, the set $\phi \left(x(t)\right)$ is the abstraction of x(t). For a given set of species X⊆M, a state $x\left(t\right)\in R_{\geq 0}^{\bar{m}}$ is an instance of X if and only if its abstraction equals X.*


The notions of abstraction and instance connect the properties of the reaction network with the properties of the reaction system. Namely, organizations represent the abstractions of all the possible stable instances:

**Theorem** **1.**
*If x is a fixed-point of the ODE Sv(x,k)=0, then the abstraction ϕ(x) is an organization [26].*


Fixed points entail the simplest dynamically stable instances of a reaction system and are crucial for determining the most important features of the dynamics of a system [27]. Thus, Theorem 1 provides a link between the long-term behavior of a reaction system and its underlying reaction network. In simple words, it proves that a necessary condition to be a fixed point is to be an organization. Moreover, in [28], Theorem 1 is extended to other stable asymptotic behaviors such as periodic orbits and limit cycles. In addition to these results, necessary conditions for the existence of adequate processes, and their relation to other theories that connect structure and dynamics such as Petri–Nets and Deficiency theory, are explored in [14,15,29], and algorithmic studies concerning the computation of the organizations of a reaction network are presented in [30,31,32,33,34].

COT does not explain whether or not the organization will reach a fixed point, or if it will persist under a small perturbation. That is why we aim in the next sections at showing what happens when perturbations are in play, by looking into the interplay between state and flux. However, it is necessary to explain first that there exist three fundamentally different types of change when speaking of a system described as a reaction network [12].

### 3.3. Three Types of Change for Perturbations, Actions, and Goals

The reaction network approach provides a suitable landscape of concepts to formalize some modern systemic notions. Take for example the notion of resilience defined as *the ability of a system to cope with change*. By “cope”, authors generally mean “to be able to maintain its qualitative identity”, and by “change” they mean “a perturbation”. However, the notion of qualitative identity, as well as the notion of perturbation, are generally referred in resilience literature in a non-formal manner.

In our approach, the qualitative identity of a system corresponds to an organization together with self-maintaining processes being applied to it. Therefore, a subnetwork of the reaction network operating in a particular manner. Once the network acquires a particular qualitative identity, we observe there are three different types of change, either for a perturbation, action, or goal. The first is a change of state, and this means increasing or decreasing the values of the coordinates of the vector x. At the level of perturbations, this represents the influence of the environment in the system. At the level of actions, this represents its dynamical evolution. At the level of goals, this represents a new end-state to be seeking for. The second is a change of the dynamical rules, so-called process change, and this means changing the set of possible processes v that can occur in the system. At the level of perturbations, this corresponds to an environmental sudden change of the parameters that rule the evolution of the system (e.g., an external increase in temperature). At the level of actions, this corresponds to a regulatory mechanism within the system (e.g., signaling mechanism that foster or inhibits the happening of certain reactions). At the level of goals, this corresponds to a new set of preferred behaviors (a change from chaotic to non-chaotic behaviour). The third is a change of the very dynamical rules, so-called structural change, and this means adding or eliminating species and/or reactions in the system. At the level of perturbations, this corresponds to an environmentally-driven arrival or disappearance of a reaction (e.g., the invasion of a new species in an ecosystem). At the level of actions, this corresponds to the creation of a new form of behavior (e.g., learning mechanisms that foster survival). At the level of goals, this corresponds to a a new preferred form structural operation (e.g., ecosystem adaptation).

In general, perturbations, actions and goals can be composed of the three types of change expressed previously. See Table 1 for a summary of these notions.

## 4. Goal Directedness Examples

In this section, we will discuss in more detail how we can recognize goal directedness modeling systems as chemical organizations. Namely, we will construct basic organizations, starting from the most simple negative feedback, and slowly building up to slightly more complex examples.

As we want to explore negative and positive feedback in COT, we should first understand these mechanisms better, to know whether a cycle is a positive or negative feedback. We say that *A* has a positive influence on *B* if more of *A* leads to more of *B*, and less of *A* leads to less of *B*. *A* has a negative influence on *B* if more of *A* brings less of *B*, and less of *A* brings more of *B*. We had a positive feedback if more of *A* brings more of *A*, thus if *A* has a positive influence on itself. We can see that this will in general be the case if there is a cycle with an even number of negative influences, for example, a combination of two positive influences (the more *A*, the more *B*, and the more *B*, the more *A*), or a combination of two negative influences (the more of *A*, the less of *B*, and the less of *B*, the more of *A*). While a negative feedback is a cycle with an odd number of negative influences, for example, a combination of a positive and a negative influence (the more of *A*, the more of *B*, and the more of *B*, the less of *A*), leading to more *A* (than a certain threshold) lowering *A* again, while less of *A* will lead to more of *A*.

To be able to speak about positive and negative influence in COT, we should first make some assumptions on the relation between the process rate and the concentrations of the reactants. That is why in the following we assume monotonic dependency, by which we mean that the process of a reaction depends on its reactants in a monotone way: the more there is of a reactant, the more (or equal) the reaction will occur. Mass action kinetics is a typical example where the later holds [35].

Then, if a reaction is producing a certain species, there is a positive influence of the reactants on that species: the more there is of a reactant *A*, the more of (product) species *B* will be produced, and less of *A* brings less production of *B*. While with consumption, the influence is negative: if *B* is being consumed by a reaction that has *A* as reactant, more of *A* will bring less of *B*, and less of *A* will lead to (relatively) more *B*. However, even in simple reaction networks, there will be a complicated combination of positive and negative influences; one reason for this being that a species has a negative influence on itself (as it triggers its consumption), often leading to some cycles with an odd number of negative influences (negative feedback), and other cycles with an even number (positive feedback). That is why it is non-trivial to see whether there will be a positive or a negative feedback, because it is unclear to see which cycle(s) will prevail. For this reason, we will work with simple examples in the following, to identify which kind of feedback prevails in these cases.

To be able to compute specifics, we will assume mass-action kinetics [35].

Mass-action kinetics specifies that the process of a reaction is proportional to the product of its reactants:(4)$$v\left[i\right]=k_{i}\prod_{j=1}^{m}x{\left[j\right]}^{a_{ij}}$$
with $k_{i}$ a constant, called a kinetic parameter. As in the general case, the change of the concentrations depend on the processes:$$\dot{x}=Sv$$

This expresses the idea that the change is the sum of what is produced (at a rate determined by the process), minus what is consumed.

We will look what happens if there is a perturbation, by which we can identify a negative feedback. Examples one to four consider state perturbations, i.e., by changing the concentrations of some species, and process perturbations by changing the kinetic parameters of the processes. The fifth example considers a structural perturbation, changing the reactions in the system. If the organization returns to the same concentration of a species, there is a negative feedback: deviations in this concentration are counteracted (end-state). However, as seen previously, there can be cases where some ratios between the concentrations remain the same, while the concentrations themselves change (end-behaviour), or when part of the structure remains stable even though there is change in concentrations and behaviour (end-structure).

### 4.1. Direct Negative Feedback of a Species on Itself

If a species *a* is consumed, it can have a negative feedback on itself: the more there is of *a*, the more it will be consumed and thus the less of it there will be. To describe this mathematically, consider following reactions:$$\begin{array}{c} r_{1}:\emptyset \to a\\ r_{2}:a\to \emptyset \end{array}$$

Then, the dynamical evolution of *a* is described by
$$\begin{array}{c} \dot{a}=k_{1}-k_{2}a \end{array}$$

This will always move to an end-state, i.e., to $\dot{a}=0$. Namely, if $\dot{a}>0$, *a* will increase, and thus $\dot{a}$ will decrease, moving closer to zero. While if $\dot{a}<0$, $\dot{a}$ increases. Therefore, we will reach a fixed point where $k_{1}-k_{2}a=0$, and thus $a=\frac{k_{1}}{k_{2}}$.

This fixed point can be seen as a goal. For state perturbations (thus changing the concentration of *a*), the end-state does not change. However, for process perturbations (changing the parameters $k_{1}$ or $k_{2}$), the end-state changes, but the behavior of *a* remains stable (towards a particular end-state). Note that in the extreme cases $k_{1}=0$ or $k_{2}=0$, we encounter a structural change because one of the reactions is eliminated from the dynamics. When $k_{1}=0$, we have that *a* will deplete so the end-state has a different structure (again a single end-state, but no process, and no structure). Whereas when $k_{2}=0$, we have that there is no fixed point to reach as the system will increase its concentration indefinitely (no end-state, no end-state oriented process, but same structure because *a* is still present). Structural perturbations in which reactions are added can induce different types of change. For simplicity, we will avoid this kind of change but study other structures.

### 4.2. Cycle of Mutual Production

The previous example assumes that the production of *a* is independent from the concentration of *a*. To investigate what happens when this is not the case, we consider a cycle of mutual production:$$\begin{array}{c} r_{1}:a\to b\\ r_{2}:b\to a \end{array}$$

Here, species have a positive influence on each other, while they have a negative influence on themselves. It is thus not directly clear what kind of dynamic will prevail. The time-evolution is described by
$$\begin{array}{c} \dot{a}=k_{2}b-k_{1}a\\ \dot{b}=k_{1}a-k_{2}b=-\dot{a} \end{array}$$

This will also move towards a fixed point: if $\dot{a}>0$, we automatically have $\dot{b}<0$, *a* increases and *b* decreases, thus $\dot{a}$ decreasing and $\dot{b}$ increasing, both moving closer to zero, and similarly for the other case. $k_{2}b-k_{1}a=0$ gives $b=\frac{k_{1}a}{k_{2}}$. The ratio between *a* and *b* will hence stabilize. The exact values *a* and *b* will take however now depend on from which values we start from. We have $0=\dot{a}+\dot{b}={\left(a+b\right)}^{\prime }$, thus a+b remains constant at any time. With these two equations, we can unambiguously calculate the concentrations of *a* and *b*.

Therefore, with state perturbations, the exact values of *a* and *b* will now change (as the constant to which their sum must be equal, will change), but the ratio between them will remain constant. For process perturbations, this ratio will change, but the sum of the concentrations will remain constant. For any state or process perturbation, the rate of the two reactions will be equal, $v_{1}=v_{2}$.

Note that a kind of negative feedback mechanism is present here, but at the level of the processes: when one process occurs more than the other, it will lower its occurrence while the other will increase, until they are equal. This also brings that the species are stable and balance each other out.

### 4.3. One Cycle with Stable Ratio Influencing Another Part, Giving Stable State

We will now build another layer on top of the previous example, where we show that the specific concentrations of *a* and *b* do not matter to bring stability to another species, only the ratio between them matters. For this, we add following reactions:$$\begin{array}{c} r_{3}:a+x\to a\\ r_{4}:b\to b+x \end{array}$$

As the consumption and production of *a* and *b* are not modified by these reactions, their dynamics will remain oriented towards an end-state. We consider thus the change in *x*:$$\begin{array}{c} \dot{x}=k_{4}b-k_{3}ax \end{array}$$
there is a same negative feedback at work as in the first example. Therefore, *x* will stabilize at
(5)$$x=\frac{k_{4}b}{k_{3}a}=\frac{k_{4}\frac{k_{1}a}{k_{2}}}{k_{3}a}=\frac{k_{4}k_{1}}{k_{2}k_{3}}$$

We see that indeed the end-state of *x* is independent of *a* and *b*. Thus, *x* will orient towards the same value even under state perturbations. Process perturbations will still alter this value when the ratio (Equation 5) changes. However, under any state or process perturbation the end state will still hold $v_{3}=v_{4}$.

### 4.4. Catalyst Influencing the Self-Maintenance and Over-Producibility

We will now look at the opposite phenomenon compared to the previous example: when the exact value of a species matters for the self-maintenance of the system. Consider the following reactions:$$\begin{array}{c} r_{1}:a+c\to 2a+c\\ r_{2}:a\to \emptyset \end{array}$$

Therefore *a* is both producing and consuming itself, while there is a catalyst *c* helping with the production. The change of *a* is defined as
$$\dot{a}=k_{1}ac-k_{2}a=\left(k_{1}c-k_{2}\right)a$$

The dynamics now depend on the sign of $\left(k_{1}c-k_{2}\right)$, which depends on the concentration of *c*. If $(k_{1}c-k_{2})=0$ (this is when $c=\frac{k_{2}}{k_{1}}$), there will be stability, *a* remains at whatever value it started of. When $(k_{1}c-k_{2})>0$ ( $c>\frac{k_{2}}{k_{1}}$), there will be overproduction of *a* because $\dot{a}>0$. With $c<\frac{k_{2}}{k_{1}}$, *a* will deplete because $\dot{a}<0$ until *a* reaches zero, so the reaction network is not self-maintaining.

Therefore, in this case both state and process perturbations can break down the organization {a,c}.

### 4.5. An Example Incorporating Structural Perturbations

First, consider the reactions of the example in Section 4.2:$$\begin{array}{c} a\to b\\ b\to a \end{array}$$

We know this system is going to stabilize in a way that a+b is constant but dependent on the total amount of *a* and *b*. If we perturb structurally this reaction network including the reactions studied in Section 4.3 as follows:$$\begin{array}{c} x+a\to x\\ y\to y+a \end{array}$$

We obtain that the resulting reaction network will stabilize in a state which is independent of the total amount of *a* and *b*.

Indeed, we have that
$$\begin{array}{c} \dot{a}=-k_{1}a+k_{2}b-k_{3}xa+k_{4}y\\ \dot{b}=k_{1}a-k_{2}b \end{array}$$

Then,
(6)$$\dot{a}+\dot{b}=-k_{3}xa+k_{4}y,$$
so setting $\dot{a}+\dot{b}=0$, we obtain $a=\frac{k_{4}y}{k_{3}x}$, implying $b=\frac{k_{1}k_{4}y}{k_{2}k_{3}x}$. Therefore, both concentrations are now independent of the total amount of *a* and *b*.

In this sense, the action of including these reactions in the system changes not only the end-state value of *a*, but also and more interestingly the behaviour of the system. Namely, the structural perturbation induces resistance of to state perturbations in a way that was not possible in the prior the structural perturbation.

## 5. Discussion

We see that in our examples there is some kind of negative feedback: something remains, some law is present to which the system returns. In some cases, the concentrations always moved to the same constant, but sometimes they only held the same ratio towards each other. In these examples, the processes always became equal to each other (but this will not necessarily be the case in other reaction networks). The specific equalities to which the system moves can be derived from the fixed point equation $\dot{x}=0$, and we explained why in the first examples, the system moved towards this fixed point (negative feedback dynamic).

We can see that the same dynamics would hold if we do not assume kinetic law, and only assume a monotonic dependency on the reactants. Because the only thing we used in our reasoning is that the process increases if the reactant of that reaction increases, and not the specific way this happens under mass-kinetics. We can for example express the first example as that when *a* increases, the process consuming *a* will increase, which will lower the production rate of *a*, moving $\dot{a}$ closer to zero.

Within a set of mutually influencing species, we saw there was some resilience: if the reaction network was self-maintaining, it remained so under the face of (state or process) perturbations. The mutual influence allows for the possibility of negative feedback: if a species is being consumed more than produced, there can be a mechanism changing this, and the fact that the reaction network has been self-maintaining, assures such a state where any species is produced as much as consumed, is possible. We saw this phenomenon at work in some basic examples, but we hypothesize that it holds more general, under the assumption of monotonic dependency of the process on the reactants, and a self-maintaining (and closed) set of mutually influencing species.

Perturbations of different kinds provide ways to modify the goals at different levels. It is of high interest to conceive evolutionary systems in which these capacities are developed across generations. This relates to different philosophical notions on goal-directed behaviour. We do not want to see a goal as merely a target that should be reached, but also consider transformability, adaptability, growth, and the constant creation of goals. The pure self-maintenance of an organization can be seen as the goal as a target: the focus here is on maintaining something. Under perturbations, the organization can change, new goals can be created, it can become more resilient precisely because it adapts. Here, the focus is on the value of change. To be goal-directed, it should however still drive its own change.

While the notion of chemical organization is an operationalization of the notion of goal, it accounts for goals at an abstract level only. Namely, goals correspond to structures (subnetworks of processes) able to persist in time. Fortunately, the reaction networks formalism has a large number of theoretical variants which advance the relation between its productivity structure and its ability to persist in time under different constrains. In particular, COT has been compared to important fields of research in reaction networks such as autocatalytic sets [13], deficiency theory [14], Petri nets [15], and elementary modes [31].

Expanding on the relations between COT and not only these reaction networks frameworks, but also other interesting frameworks, such as structural kinetic modeling [36] and pathway analysis [37], and others [38], allows for a deeper description of the notion of goal and suggest the existence of different classes of goals, each of them operationalized differently as shown in Table 1.

It is future work to relate our approach to these and other concepts, and this will bring us closer to an answer to the big question of the origin of life. To do so, we propose investigating classes of reaction networks having certain cyclic or productive properties that have been related to the emergence of life. Among them autocatalytic sets [9], reversible reaction networks [39], and networks with specific properties leading to persistence such as those explored by deficiency theory [23,40]. Another interesting issue to explore is to understand how modularity of reaction networks [41] is related to our notion of goal. Namely, a goal might be reached by the parallel (or weakly coupled) work of different units. This can be represented using reaction networks by dynamical decomposition methods which allow to disentangle the dynamics of the network into independent parts [32,42].

It is also important to develop an entropy measurement in COT, as life can be seen as a reduction in entropy, far-from-equilibrium. As a goal was defined as “something that remains”, a goal in some cases implies entropy reduction. Namely, when the dynamical behaviour of the reaction network counteracts the diffusive evolution towards the statistically most likely state, we see that the system self-maintains in a low entropy situation. The latter is a sign of the emergence of life and should be related to our notion of goal as well. Further, we could distinguish between external and internal perturbations (macroscopic fluctuations and internal reaction noise). Internal perturbations can be thought of as actions the system apply on itself, while external perturbations depend on the environment. However, both can help or prevent reaching a goal. The interplay between these perturbations is also a future line of exploration. Another interesting extension is to explore more complex reaction networks where, different to our examples, the system tends to a state where the process rates are not the same for all reactions. These less trivial stable states indicate some sort of complex reversibility, because there is as much produced as consumed of each species, but not all reactions need to be triggered at the same rate. When this effect is amplified we see the emergence of timescales for different reactions, and this indicates the emergence of hierarchical behaviour in the reaction network [43].

## 6. Conclusions

We have proposed a framework to define and investigate how to define goal orientedness using reaction networks as a representational language [12] in which cybernetic mechanisms come into play to create circular causalities based on feedback mechanisms [3]. We defined and operationalized three fundamental notions to comprehend the notion of goal directedness: perturbation, action, and goal. These three notions depend directly on three types of change that can be defined within the reaction network formalism [32]: state, process, and structural change. We presented some examples that illustrate our notion of goal-directedness and discussed the forms in which this research program can be extended.

An important topic for further consideration is that besides perturbations, we can also look at the interactions between goals, how different goals can coordinate, and what happens when basic goal-directed structures, negative feedbacks, mechanisms or organizations interact. When two or more organizations interact, a new reaction network is formed. As the new reaction network can have reactions that are not in any of the former organizations, the union of two (or more) organizations may or may not be an organization [34]. These new reactions can trigger new products, and thus the union is not necessarily closed. Then, synergies exist between the organizations, and they can evolve in an organization that is bigger and richer in its dynamics or cybernetic mechanisms than the union.

## Figures and Tables

**Figure 1 entropy-23-01039-f001:**
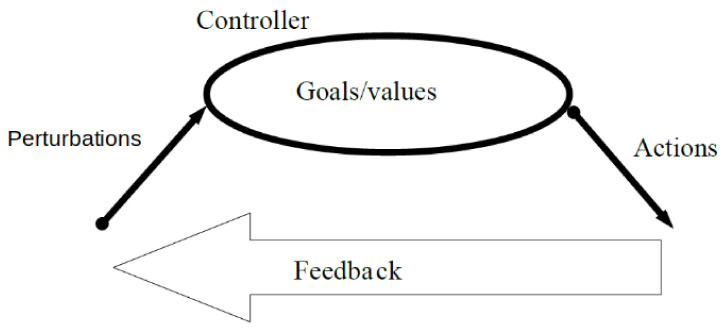
Interplay between goals, actions and perturbations. Feedback mechanisms emerge to modify actions in response to perturbations.

**Table 1 entropy-23-01039-t001:** Interpretation of the types of change for perturbations, actions and goals.

Change	Perturbation (in Phase Space)	Action	Goal
State	another point	causal effect	end-state(s)
Process	possible directions	dynamic control	end-behavior
Structural	dimension	rule control	end-structure

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
