# Peer review of "Goal Directedness, Chemical Organizations, and Cybernetic Mechanisms"

_entropy, 2021, doi:10.3390/e23081039_

Round 1
Reviewer 1 Report
This manuscript attempts to give a first thought of how so-called goal-directed systems could be rationalized in terms of chemical reaction networks. The authors present and analyze a number of very simple sets of chemical processes as examples to underline this idea. In this respect, it is a conceptual paper, which is meant as starting point for future studies of perhaps more complex networks, as stated by the authors themselves. From my view, this manuscript is technically sound, well written and gives a nice inspiration.
On the other hand, it would be interesting if the authors could include and discuss in their outlook for future attempts some additional basic aspects that perhaps have to be considered for COT, such as when talking about closed cycles to remind microscopic reversibility, to discuss autocatalytic networks and autocatalytic orders, to take into account more specifically the entropy production in far-from-equilibrium scenarios, to distinguish between the effects of macroscopic fluctuations and internal reaction noise, the consideration of open systems. All these points could be developed using similar simplified chemical processes in future works and are possibly more enlightening to relate the task to the big question about the origins of life.
Author Response
Dear Reviewer,
Thank you very much for your suggestions. As proposed, we elaborated further in the discussion on how to approach classes of more complex reaction networks such as reversible, or autocatalytic RNs. Moreover, we expanded a little bit as well on the relation between life, entropy and goal. We also elaborated on various other aspects that deserve further exploration.
Though, we proposed all that as future work in the discussion as the aim of the article here is to discuss some basic examples and their relation to feedback mechanisms in the context of goal-directedness. You can find such paragraphs, and all other changes in blue text.
Thank you very much.
Reviewer 2 Report
Dear Editor, Dear Authors,
the manuscript "Goal-directedness, chemical organizations and cybernetic mechanisms" (entropy-1287078) submitted to Entropy show an interesting framework by means of goal-directedness, reaction networks and chemical organizaton theory. This manuscript is very well written and its contents is clearly presented. I recommend publication after refining the manuscript according to these two minor comments:
Line 6
build introduce ?
Line 8
Please define COT above
Author Response
Dear Reviewer,
Thank you very much. We proceeded with the minor changes as requested.
All changes with respect to the previous version can be tracked by looking at the blue text.